

# Hypoxia in mangroves: occurrence and impact on valuable tropical fish habitat

Alexia Dubuc[1,2], Ronald Baker[3], Cyril Marchand[4], Nathan J. Waltham[1,2], Marcus Sheaves[1,2]

[1]College of Science and Engineering, James Cook University, Townsville, 4810, Australia
5 [2]TropWATER, Townsville, 4810, Australia
[3]Department of Marine Sciences, University of South Alabama, Dauphin Island Sea Lab, Alabama, 36528, United States of America
[4]Institute of Exact and Applied Sciences, University of New Caledonia, Noumea, 98800, New Caledonia

10 *Correspondence to*: Alexia Dubuc (alexia.dubuc1@my.jcu.edu.au)

**Abstract.** Intertidal mangrove forests are harsh environments that can naturally experience hypoxia in association with low tide. However, we know relatively little about dissolved oxygen (DO) fluctuations and DO-induced responses by fish, although DO is a fundamental water quality parameter. This study examines DO as a potential factor regulating the utilisation of intertidal mangrove forests by fish, and consequently their value. We deployed underwater video cameras, 15 coupled with DO and depth loggers, in a mangrove forest to record changes in fish assemblages in response to tidal variations in DO and other associated environmental parameters. Our results indicate that DO underwent extreme tidal fluctuations, reaching levels as low as 14 % saturation. As DO was identified as a significant factor to explain variability in fish assemblage composition, we further investigated fish responses to DO fluctuations. Higher taxonomic richness and frequencies of occurrence were observed once DO reached 70-80 % saturation. More detailed examination revealed species-20 specific responses. Three distinct patterns of mangrove utilisation in response to DO were identified, driven by apparent taxa's behavioural DO avoidance thresholds. Most taxa did not display any behavioural avoidance, including presence at the lowest DO levels, while other taxa were not observed either below 50-60 % saturation, or below 70-80 % saturation. This implies that tidal migrations, often observed in intertidal environments, could be the result of differential DO tolerances, and not simply initiated by changes in water depth. Taxa remaining in the mangrove forest even at low DO were on average 25 more frequently observed than the other taxa, and were mostly species commonly associated with mangrove habitats. This suggests that being adapted to withstand low DO might be an important condition to use mangrove habitats extensively. The need of being tolerant to low DO could constrain fish utilisation and explain the relatively low species richness often observed in other intertidal mangrove forests.

## 1 Introduction

30 Mangrove forests are recognised as important habitats for fish (Robertson and Duke, 1990; Nagelkerken et al., 2002; Nagelkerken et al., 2008). However, their value has been shown to be heterogenous and influenced by local environmental



factors influencing mangrove forests accessibility and suitability (Faunce and Serafy, 2006; Bradley et al., 2019). Mangroves can be challenging habitats, especially in regions where they are subjected to tide (Unsworth et al., 2007; Olds et al., 2012), as tidal variation generates a range of constraints. Mangrove forests generally become only accessible for short periods while flooded at high tide, and the decrease in water depth as the tide ebbs leads to eventual drainage of the forest (Sheaves, 2005; Baker et al., 2015). Tidal variation also induces short-term changes in environmental conditions such as salinity, temperature, water depth, turbidity, light, and dissolved oxygen (DO), that can lead to a temporarily unsuitable habitat for fish utilisation (Davis, 1988; Rountree and Able, 2007; Brady and Targett, 2013; Dubuc et al., 2017; Mattone and Sheaves, 2017).

Several studies have demonstrated that fish undertake regular migrations in intertidal mangrove forests. Migration patterns have been shown to be species-specific and influenced by tide (Laroche et al., 1997; Krumme, 2004; Ellis and Bell, 2008; Meynecke et al., 2008; Sheaves et al., 2016; Dubuc et al., 2019). Tidal migrations indicate that fish respond to one or several factors varying with tide. However, there is still uncertainty on what environmental factors induce these tidal migrations. The factors driving these species-specific tidal migrations could be changing water depth (Bretsch and Allen, 2006; Ellis and Bell, 2008; Reis-Filho et al., 2016), or alternatively the result of active avoidance to adverse changes in water quality.

A key factor determining water quality and that can change drastically across tide is DO. DO is crucial for all aerobic organisms, including fish (Driedzic and Hochachka, 1978; Falkowski and Raven, 1997). However, DO availability varies extremely over the tidal- and diel-cycle in mangrove habitats, reaching levels that can lead to physiological stress (Knight et al., 2013; Dubuc et al., 2017; Gedan et al., 2017; Mattone and Sheaves, 2017). Consequently, it is likely that some fish species respond to changes in DO by undertaking tidal migrations, or by avoiding mangrove forests permanently to prevent the adverse effects following exposure to low DO. Despite the importance of DO and its extreme variability in shallow-water environments, our understanding on how DO fluctuations shape patterns of fish utilisation on a tidal and diel scale is limited (Davis, 1988; Smith and Able, 2003; Rountree and Able, 2007).

DO is maybe the most complex and variable parameter to study, as it is influenced by multiple interacting biotic and abiotic parameters at a range of spatial and temporal scales (Buffoni and Cappelletti, 1999; Diaz and Rosenberg, 2008; Nezlin et al., 2009). However, DO is partially predictable, with the lowest DO levels occurring at night or dawn at low tide, following nighttime respiration, while maximum levels are recorded in the afternoon at high tide, following autotrophic production (Kenney et al., 1988; Mazda et al., 1990; D'Avanzo and Kremer, 1994; Tyler et al., 2009). This diel pattern gives part of the answer of when fish could be able to use mangrove habitats without being exposed to high risks of low DO. However, fish species have developed physiological and behavioural adaptation strategies (Kramer, 1987; Breitburg, 1994; Diaz and Rosenberg, 1995), leading to species-specific hypoxia tolerances (Vaquer-Sunyer and Duarte, 2008). Consequently, these adaptations could result in species-specific tidal migrations, as species highly tolerant to hypoxia would be adapted to use more often and remain longer in mangrove forests, compared to other less tolerant species that would be restricted to access mangrove forests at higher DO levels.





Although it is known that mangrove forests experience natural low DO in some locations (Knight et al., 2013; Mattone and Sheaves, 2017), it is unknown how general this phenomenon is, and what are the potential consequences on fish populations. Understanding DO dynamics and the impacts on fish utilisation and value of highly productive habitats such as mangroves is crucial, especially in the context of global ocean deoxygenation (Diaz and Rosenberg, 2008; Breitburg et al.,
2018). Ocean deoxygenation is mainly due to the increase of human activities along the coastlines during the past 50 years (Vaquer-Sunyer and Duarte, 2008), implying that mangrove forests are especially vulnerable to anthropogenic deoxygenation due to their location along the coasts. By addressing the gaps of knowledge around hypoxia in mangrove forests, managers would be in a stronger position to implement adequate action plans to limit the impact of hypoxia that is predicted to worsen in the coming years (Breitburg et al., 2018).

This study examines the impact of DO fluctuations on the utilisation of mangrove forests by fish in a mangrove-coral reef seascape in the Indo West Pacific (IWP). We assessed how fish utilisation changes across tidally varying DO levels, and we determined the relative importance of DO, depth, lunar phase (neap vs spring), location within the mangrove forest (edge vs in-forest), time of day, and tide direction (flooding vs ebbing) in explaining variations in fish assemblages. To address this aim, we used unbaited underwater video cameras, simultaneously deployed at dawn and mid-afternoon on the
edge and 5 m inside a mangrove forest, coupled with high frequency DO and depth loggers. The study site was located in an IWP mangrove-coral reef seascape experiencing a microtidal regime.

## 2 Materials and Methods

### 2.1 Study site

The study was conducted in a semi-enclosed lagoon (1.2 km long, 60 m wide, 1-2 m depth) located in Bourake, South
Province of New Caledonia (21° 56.971S, 165° 59.481E; Fig. 1). The system comprises a 2.5 km2 mangrove forest dominated by *Rhizophora stylosa* on the seaward edge and *Avicennia marina* on the landward margin. A channel (20-70 m wide, 2-6 m depth, 700 m long) bisects the mangrove forest and connects the semi-enclosed lagoon to the coastal waters of Pritzbuer Bay (~ 20 km2). The channel comprises two sheltered inlets (approximately 0.01 km² each), and a shallow (1-2 m depth) coral reef platform that extends from the middle of the channel to the edge of the mangrove forest. New Caledonia is
an archipelago located in the South West Pacific, around 1500 km east of Australia. It is characterised by a semi-arid to tropical climate with annual total rainfall of 1000 mm, and a mixed semi-diurnal microtidal regime (maximum 1.8 m tidal range). The study system receives little freshwater inflow with no defined drainages.

**Figure 1: Map of the study system in Bouraké, South Province of New Caledonia. The 9 study sites sampled from 21 February to 1
March 2017 are indicated by their corresponding numbers. Light grey areas represent mangrove forest, dark grey areas represent mainland, and white areas represent water.**



## 2.2 Data collection

Nine sites were selected on an inland/offshore gradient along the channel (Fig. 1). Sites 1 to 8 were 4 paired sites, with odd site numbers located on the mangrove forest edge (defined as the boundary between mangrove prop-roots and bare substrate) and even site numbers located 5 m inside the mangrove forest. Site 9 was located on the edge of scattered mangrove trees

growing on the reef platform of the innermost inlet and was considered as an edge site.

Fish assemblages were examined at the sites using unbaited underwater video cameras (UVCs). UVCs were deployed at dawn until the battery was discharged (around 2.5 h) and again mid-afternoon, during neap (21 to 23 February 2017) and spring tides (28 February to 1 March 2017), simultaneously at the 9 sites. This sampling design was applied to capture fish assemblages as close as possible to the expected lowest daily DO levels (dawn), and the expected highest DO

levels (mid-afternoon; Dubuc et al., 2017). Cameras were positioned around 7 cm above the substrate, facing towards the channel. A marker was placed 0.5 m in front of the camera lens as a visibility indicator to ensure all videos had a minimum visibility of 0.5 m. Visibility was relatively consistent during the sampling period, and fish could be identified confidently up to approximately 2 m from the UVCs in all videos.

Over this study, we examined the effects of: tidal factors (depth, lunar phase (spring vs neap) and tide direction

(flooding vs ebbing)) related to habitat accessibility; DO, temperature and salinity related to habitat suitability; and two different components of the mangrove forest (edge and in-forest) related to the nature of mangrove habitats. Between 21 February and 1 March 2017, near-bottom (~ 5 cm above the sediment) DO (% saturation) and water temperature (°C) were measured every 15 minutes at each site using calibrated multi-parameter loggers (YSI Pro ODO model (accuracy ± 1 % saturation). A depth logger (In-Situ Inc. Rugged Troll 100 model) was coupled with each multi-parameter logger to measure

water depth (cm) every 15 minutes. Salinity was measured every 15 minutes from 21 to 23 February, and between 28 February to 1 March 2017 using another multi-parameter logger (YSI 6920 V2-2) positioned at site 5. Tidal range was obtained from the SHOM website (SHOM, 2017).

## 2.3 Data extraction from videos

Methodological details to extract data from the videos are reported in Dubuc et al. (2019). Briefly, as considerable time is

required to process videos, we subsampled the acquired recordings. One neap tide and one spring tide sampling were randomly selected for processing. Five sites from the second neap and spring tide sampling were also processed so one randomly selected replicate on the reef platform (site 9), and two replicates of randomly selected paired sites, not located on the reef platform, were acquired (sites 5-8). Videos were viewed using VLC and subdivided in 5-min intervals to follow the temporal variations in fish assemblages. All taxa observed in each 5-min interval were identified and recorded. Only

presence/absence data were recorded to avoid biases induced by count data when using UVCs (Sheaves et al., 2016). Fish were identified to the lowest possible taxonomic level, with all fish identifications validated by two additional experts. For





each 5-min interval video, information about depth, DO, time of day, lunar phase (neap vs spring), habitat (edge vs in-forest), and tide direction (flooding vs ebbing) was recorded.

## 2.4 Data analysis

To graphically investigate temporal dynamics of DO on the edge and in-forest, and covariance with depth and temperature,
cubic spline smoothers were fitted to the 3 time series using R. DO residuals were graphically added to emphasise extreme DO levels. A Kendall's correlation test (as DO did not follow a normal distribution) was used to determine whether patterns of change in DO were significantly correlated between edge and in-forest sites. Cumulative DO frequency curves (Dubuc et al., 2017) were plotted for each site to highlight differences in spatial and temporal dynamics.

       Following the methodology described in Dubuc et al. (2019), an index depending on observation per unit effort
(OPUE) was used to calculate frequencies of occurrence for each taxa (the total number of 5-min intervals in which a taxon was observed in one sample unit was divided by the total number of 5-min intervals recorded for the same sample unit). We acknowledge the existence of non-independence issue created by subsampling videos in 5-min intervals. Indeed, this can potentially lead to the count of the same individual fish in sequential time windows. However, the objective here was to characterise environmental conditions suitable for the utilisation of mangrove habitats through time by different taxa.
Therefore, we assumed that if an individual of a taxon was present (no matter if it was the same individual or another one to any recorded in previous 5-min intervals), then conditions were suitable. Frequencies of occurrence were first calculated per site. Only taxa with a frequency of occurrence ≥ 0.05 on at least one site were retained for analyses (hereafter referred to as "common taxa").

       We hypothesised that DO is an important factor in explaining the presence of fish taxa. To test this hypothesis, a
random forest (RF) model (Breiman, 2001) was used to quantify the relative importance of DO and the other measured environmental factors, and identify how well the combination of the selected environmental factors predicted fish taxonomic richness. This machine learning algorithm permits analysis of data that do not meet the requirements of normality and homoscedasticity required for approaches such as general linear models, and include repeated measurements (Mercier et al., 2011). The taxonomic richness was determined for each 5-min interval recorded (1434 5-min intervals in total). The dataset
was then split in two to obtain a training dataset (875 5-min intervals obtained from the two entire sampling days processed) to build the RF model, and a test dataset (559 5-min intervals obtained from the 5 replicate sites processed) to test the robustness of the model at predicting taxonomic richness. The RF model, consisting of 1000 regression trees generated using 2 predictors at each split (default value), was created to predict taxonomic richness, with "Habitat" (edge vs in-forest), "Depth", "DO", "Lunar phase" (neap vs spring), "Time of day" (morning vs afternoon), and "Tide direction" (flooding vs
ebbing) as predictors. The out-of-bag error estimate was used to validate the model. The increase mean-square error was calculated to determine the variable importance in predicting taxonomic richness. The model was then run on the test dataset to generate a confusion matrix. From the confusion matrix, the percentage of cases when the model was able to predict the exact taxonomic richness observed was calculated, as well as the percentage of cases where the model was able to predict the





taxonomic richness observed at ± 1 taxon. All RF model-related analyses were conducted using the 'randomForest' package in R (Breiman, 2001). As a RF model is built from many classification trees, it is not accurate to draw a single tree from this model. Therefore, a univariate classification tree analysis was carried out on the training dataset with the same variables as the RF, and the tree obtained from this analysis was used to visually interpret the RF model. The univariate classification tree

analysis was performed using the package "party" in R (Hothorn et al., 2010).

After quantifying the importance of DO, the goal was to understand when fish initiated responses to DO. Each 5-min video interval was allocated to a DO % saturation class in 10 % intervals (from 30-40 % saturation to 100-110 % saturation) according to the DO level recorded. The frequency of occurrence of each common taxon per class of DO was then calculated to investigate whether intensity of utilisation varied in response to DO. A General Additive Mixed Model

(GAMM) was built with log10 (X + 1) transformed frequencies of occurrence of each common taxon as the response variable, "DO" as a smooth term, and "Habitat" (edge vs in-forest) as a parametric term, using a Gaussian distribution and an identity link function. The model was built using the package "mgcv" in R (Wood, 2007). Frequencies were log10 (X + 1) transformed to reduce the impact of extreme values and improve visualisation. The frequencies of occurrence of each common taxon across DO were also plotted individually using a LOESS curve. Patterns were then investigated individually

and visually grouped by similarity of mangrove utilisation in response to DO. No grouping was imposed, and visualisation of the data identified three common patterns of mangrove utilisation across DO among all common taxa. These three patterns were based on distinct preferences for DO with taxa recorded from 30 to 110 % saturation, taxa recorded from 50-110 % saturation, and taxa recorded from 70 to 110 % saturation.

Taxa observed in mangrove habitats even at low DO (from 30 to 110 % saturation) may indicate that these fish are

well adapted to use mangrove habitats extensively and therefore expected to be observed more frequently than the other taxa. To test this hypothesis, overall frequencies of occurrence were calculated for each common taxon by dividing the total number of 5-min intervals in which a taxon was observed by the total number of 5-min intervals recorded during the study at the DO range corresponding to that taxon's assigned pattern of utilisation (30-110 % saturation; 50-110 % saturation; 70-110 % saturation). Following this methodology allows to calculate frequencies of occurrence according to the effective sample

size; it therefore overcomes the unbalanced sampling effort as species recorded across the entire range of DO would automatically be more frequent than species only recorded from 70-110 % saturation as it represents a smaller proportion of the sample size. Species-specific overall frequencies of occurrence were then plotted by type of patterns of utilisation assigned using boxplots. To test for differences in overall frequencies of occurrence among the different types of patterns of utilisation, a Kruskal-Wallis test followed by a post-hoc Dunn test were performed (data did not follow a normal

distribution).



## 3. Results

DO was highly variable at the mangrove study sites in Bourake (Fig. 2). DO reached levels as low as 14 % saturation at night low neap tides during the entire logging period, and as low as 35 % saturation during the morning hours that coincided with low spring tides while UVCs were deployed (Fig. 2; Table 1). DO closely followed the diel- and tidal cycles, with daily

maximum levels recorded during the afternoon high tide, and minimum levels during the night or early morning low tide. Temperatures followed a typical diel cycle, peaking during the late afternoon and declining at night reaching minimum levels in the early morning hours. Salinity was relatively constant during the study, ranging between 32.1 and 34.9.

**Table 1: Summary of the environmental factors during the study period.**

**Figure 2: Cubic spline smoothers for dissolved oxygen (DO), depth and temperature. Data are from 21 February to 25 February 2017 and from 27 February to 1 March 2017. For DO, edge sites are represented by the blue smoother and black points, and in-forest sites by the grey smoother and red points. For the other factors, edge sites are represented by the black smoothers and in-forest sites by the grey smoothers. Shaded areas represent sunset to sunrise. Each red box represents an underwater camera**
**sampling.**

Temporal dynamics in DO were significantly correlated between in-forest and edge habitats (p < 0.0001; r = 0.95; Fig. 2; Fig. A1). DO minima and maxima were also similar between edge and in-forest sites (Table 1). Most DO levels were between 70 and 80 % saturation, nevertheless, DO levels were equal or below 50 % saturation (adopted threshold for

hypoxia; Breitburg, 2002; Dubuc et al., 2017) for around 11 % of the time inside the forest and 21 % on the edge (Fig. 3). Mean and minimum DO levels were lower during neap tides than spring tides for both edge and in-forest sites (Table 1). The duration of low DO tended to increase with distance from the mouth of the channel, with DO at or below 50 % saturation 4 % of the time at the in-forest site closest to the channel entrance (site 2), and 14 % of the time at the in-forest site furthest from the channel entrance (site 8; Fig. 3).

**Figure 3: Site-specific cumulative DO frequencies. Each colour represents a paired site (edge and in-forest), and edge sites are represented by solid coloured lines and in-forest sites by dashed coloured lines. The solid black line indicates the mean cumulative DO frequencies across edge sites and the dashed one the mean cumulative DO frequencies across in-forest sites. The frequency of hypoxia (DO ≤ 50 % saturation) at sites 2 and 8 is indicated to help read the figure.**

Fifty-six video deployments were processed (totalling more than 118 h of video). Seventy-two taxa from 29 families were recorded, with 36 common taxa (frequency of occurrence ≥ 0.05) retained for further statistical analyses (Table 2). The full list of taxa identified is provided in Table A1.

**Table 2: The 36 common fish taxa identified by underwater video cameras at Bouraké, New Caledonia.**





We used a RF model to assess the relative importance of several environmental factors in determining taxonomic richness. The robustness of the model in predicting taxonomic richness at this location was also tested. The RF model consisted of 875 5-min intervals and 6 independent environmental factors. It explained 50.2 % of the total variance in taxonomic richness. "Depth" was the most important factor for predicting taxonomic richness with its exclusion from the model increasing the

mean-square error (MSE) by more than 61 % (Fig. 4a). "Lunar phase", "DO" and "Habitat" were also important predictors of taxonomic richness (between 45 and 55 % increase in MSE). "Time of day" and "Tide direction" were of less importance but still accounted for an increase in MSE of more than 35 %. The RF model successfully predicted the exact taxonomic richness observed on the replicate sites for 23 % of the 5-min intervals recorded, and for 60 % of them at ± 1 taxon (Table S1). As DO and depth were highly correlated, a RF model was built only with "Depth", and then only with "DO" to test for

the effect of multicollinearity on their relative importance. In both cases, the total variability explained by the model was much lower (33.42 % with only "Depth" included and 26.36 % with only "DO" included) than when "Depth" and "DO" were both included (50.22 %). The univariate tree corroborated the results of the RF in terms of variable importance, and proved to be an effective way of getting a visual interpretation of the RF (Fig. 4b). The taxonomic richness was the lowest at in-forest sites, and on the edge during spring tides. Conversely, taxonomic richness was the highest when water depth was

the deepest during neap tides, and when DO was greater than 83 % saturation.

**Figure 4: Importance of different environmental factors in explaining variations in taxonomic richness. (a) Random forest importance plot. Importance plot was obtained from a random forest model built with Site, Depth (cm), Lunar phase (neap vs spring), DO: Dissolved oxygen (% saturation), Time: Time of day (morning vs afternoon), Tide dir: Tide direction (flooding vs**

**ebbing) and Habitat (in-forest vs edge) as predictors for taxonomic richness. (b) Univariate classification tree. The tree was built using the same variables and provides a visual interpretation of the random forest model. Numbers in the boxes in each terminal leaf represent the average taxonomic richness, the number of 5-min intervals (n), and the total % of 5-min intervals that n represents.**

The RF model showed that DO was an important factor in explaining variations in taxonomic richness. We therefore further investigated fish responses to DO. Log10 transformed frequencies of occurrence of all taxa combined varied significantly across DO (GAMM: F = 3.693; p = 0.0166) and differ between habitat (GAMM: F = 11.48; p < 0.0001). On average, frequencies of occurrence were highest once DO reached 70-80 % saturation (Fig. 5a and b). The spread of the frequencies of occurrence around the median was also substantially reduced once DO was between 70 and 110 % saturation, indicating

that taxa were more equally frequent, whereas at low DO levels only a few taxa were abundant, with the rest rarely observed, or absent entirely. The patterns of utilisation across DO intervals differed between the in-forest and edge habitat (Fig. 5a and b). Although, on average, the highest frequencies of occurrence were recorded once DO reached 70-80 % saturation for both habitats, there were larger disparities between taxa at edge sites, with some being frequently observed at low DO and some being rarely observed, or absent entirely until DO reached 70-80 % saturation (Fig. 5a), while at in-forest sites, frequencies

of occurrence were more stable across DO (Fig. 5b).



**Figure 5: Variation in frequencies of occurrence of fish across DO class. Frequencies of occurrence were log₁₀ transformed. Each data point used to draw the boxplots represents the frequency of occurrence of one common taxon during a specific DO class. The blue line represents the GAMM model fitted with DO as the smooth term using a Gaussian distribution and an identity link function for (a) Edge sites; and (b) In-forest sites. Shaded areas represent the confidence interval at 95 %.**

Disparities in frequencies of occurrence between taxa were explained as fish appeared to respond differently to DO variations. We identified 3 distinct types of patterns of mangrove utilisation across DO while investigating species-specific variations in frequencies of occurrence across DO: 1) Pattern 1: "High tolerance" – these taxa (19 taxa) were recorded across the entire range of DO (30-110 % saturation) and were usually known to use mangrove habitats extensively (Fig. 6; Table 2; Fig. A2a and b); 2) Pattern 2: "Medium tolerance" – these taxa (7 taxa) were not observed once DO was below 50-60 % saturation and were also usually known to use mangrove habitats extensively (Fig. 6; Table 2; Fig. A2c); and 3) Pattern 3: "Low tolerance" – these taxa (10 taxa) were not observed once DO was below 70-80 % saturation and were usually reef-associated taxa (Fig. 6; Table 2; Fig. A2d). Figure 6 only shows one example of taxa per type of patterns, however, all the species-specific patterns are provided in figures A2a, b, c and d.

**Figure 6: The 3 common patterns of mangrove utilisation across DO identified. Each LOESS curve represents one example of taxa per type of patterns of mangrove utilisation across DO: Pattern 1: "High tolerance" represented by taxon *Fibramia lateralis*; 2) Pattern 2: "Medium tolerance" represented by taxon *Acanthopagrus* sp.; 3) Pattern 3: "Low tolerance" represented by taxon *Heniochus acuminatus*. LOESS curves were built with the log₁₀ transformed frequencies of occurrence.**

The type of patterns followed by a taxon appeared to significantly influence its overall frequency of occurrence (Fig. 7; Kruskal-Wallis: $\chi^2 = 9.8757$; $p < 0.01$). Taxa following a "High tolerance" pattern were on average significantly more frequently observed than taxa following a "Low tolerance" pattern (Dunn test: $p < 0.01$). Overall frequencies of occurrence of taxa following a "Medium tolerance" pattern were intermediate but not significantly different than "High tolerance" taxa (Dunn test: $p > 0.5$) or "Low tolerance" taxa (Dunn test: $p > 0.1$).

**Figure 7: Relationship between frequencies of occurrence and type of patterns followed. Overall frequencies of occurrence were calculated for each common taxon at the DO range corresponding to that taxon's assigned pattern of utilisation. Differential letters above boxes denote statistically different means of frequency of occurrence among types of patterns of utilisation (Dunn test: $p < 0.05$).**

## 4. Discussion

### 4.1 Tidal migrations: stranding or hypoxia?

Fish assemblages were shown to be highly variable over time and space in the study area (Dubuc et al., 2019). About half of this variability was explained by multiple environmental factors among which depth, DO, lunar phase, and location within the mangrove forest (edge or in-forest) were the most important. The main trend identified among the temporal variability in



fish assemblages occurred at a tidal scale (Dubuc et al., 2019). This highlights that fish were responding to one or several factors covarying with tide. Tidal variations in fish assemblages are common in intertidal environments (Laroche et al., 1997; Ellis and Bell, 2008; Becker et al., 2012), however, the factors responsible for their occurrence have rarely been investigated. There was a high collinearity between depth and DO as both varied across the tidal cycle, and these two factors

were greatly important to explain variations in fish assemblages. Consequently, it is likely that depth and DO play an essential role in triggering tidal migrations. Previous studies have shown that fish can respond to both water depth and DO changes (Wannamaker and Rice, 2000; Bretsch and Allen, 2006; Johnston and Sheaves, 2007; Rountree and Able, 2007; Ellis and Bell, 2008; Brady and Targett, 2013), emphasising the idea that fish could be using depth and DO interchangeably as cues to initiate tidal migrations, a trigger that might be dependent on the perceived upcoming risk (stranding or hypoxia).

Depth becomes limiting when fish cannot safely access the area because it is too shallow, with associated risk of stranding. However, many taxa, including small sized species, avoided mangrove habitats even when they potentially had enough water (Dubuc et al., 2019). On the other hand, changes in DO can rapidly impair fish fitness (Chabot and Claireaux, 2008; Vaquer-Sunyer and Duarte, 2008). Indeed, in aquatic environments, DO is considered as the primary limiting factor (Fry, 1971; Claireaux and Chabot, 2016) as it is naturally scarcer than in the atmosphere (Diaz, 2001), making it a perpetual

challenge for fish to access available oxygen in the water. In the mangrove forest examined here, changes in DO across tide were extreme, with up to 80 % loss during one tidal period (high to low), supporting the notion that DO could be an important constraint for fish to access mangrove habitats, even when depth is suitable. Considering the relevance of both factors, and the fact that the risk of stranding and hypoxia are concomitant, it is likely that fish are adapted to respond to either depth or DO, depending on which one becomes limiting first, and this may vary among taxa.

The hypothesis that fish can interchangeably respond and tolerate adverse depth and DO conditions was supported by the fact that all taxa that access mangrove habitats at low depth in Bourake (Pattern 3: "Low-depth users"; Dubuc et al., 2019) were all following a "High tolerance" pattern here in response to DO, indicating that they were able to tolerate low depth as well as low DO. The effects of depth and DO might well be mostly confounded as DO fluctuations overall follow depth, however, DO amplitude (difference between minimum and maximum levels) depends on many interacting factors

including weather (Tyler et al., 2009), local geomorphology or biological and chemical activities (Mazda et al., 1990; Peña et al., 2010). Therefore, minimum and maximum DO levels for a same depth can differ and vary in complex spatial and temporal scales, independent from the tidal scale, probably explaining why depth and DO were both highlighted as important factors. These results emphasise the importance to understand the DO dynamics and its impacts on fish to comprehend how mangrove forests are being used.

**4.2 Tidal-induced dissolved oxygen variations**

During this study, we hypothesised that DO could be an important limiting factor for fish utilising intertidal mangrove forests, and our findings support this hypothesis. However, the associated risk of hypoxia in the study system was still to be tested. Diel-hypoxia conditions observed in other mangrove systems (Knight et al., 2013; Dubuc et al., 2017; Gedan et al.,



2017; Mattone and Sheaves, 2017) was also a seemingly common condition in Bourake. DO showed extreme and rapid fluctuations with the diel and tidal cycles. Low DO was recorded daily during nighttime when the tide was ebbing, reaching levels that can compromise fish fitness (Rogers et al., 2016). It is likely that hypoxia is a common condition of intertidal mangrove forests due to the mineralisation of a large amount of organic matter produced by mangrove trees, responsible for

a high consumption of oxygen by bacteria (Alongi et al., 2004; Dittmar et al., 2006), but also due to the exchange of porewater between sediments and water column, known as "tidal pumping" (Li et al., 2009; Gleeson et al., 2013; Call et al., 2015; Leopold et al., 2017). Briefly, at each flooding tide, water infiltrates intertidal sediments and then drains back to the water column during the next ebbing tide. While in the sediments, water becomes enriched in reduced compounds such as $NH_3$, $H_2S$, $FeS_2$, resulting in water acidification and deoxygenation (Marchand et al., 2011). As porewater accumulates in

the water column throughout ebbing tide (Bouillon et al., 2007), it drives extreme drops of oxygen usually observed at low tide. Connectivity with the Pritzbuer Bay was crucial here in this mangrove-coral semi-enclosed lagoon as the flooding tide presumably brings oceanic water that is more saturated, replenishing DO levels. During spring tides, higher DO levels were recorded, probably driven by higher water renewal compared to neap tides.

### 4.3 Species-specific responses to DO variations

Fish significantly responded to DO variations, with taxonomic richness and average frequencies of occurrence higher and more consistent once DO reached 70-80 % saturation. This result indicated that DO levels reached during the study were probably low enough to cause harmful effects, and therefore, many taxa responded by temporarily avoiding the area. Apparent behavioural avoidance thresholds observed were species-specific and were initiated at different DO levels, potentially driven by differential tolerances to low DO (Claireaux and Chabot, 2016). Three main types of patterns of

mangrove utilisation were identified driven by taxa's behavioural avoidance thresholds. Most taxa did not display any behavioural avoidance ("High tolerance" pattern), and some of these taxa even reached their maximum frequency of occurrence at the lowest DO levels recorded (30-40 % saturation). Most of these taxa are known to commonly use mangrove habitats such as *L. argentimaculatus*, Mugilidae spp., and Gobiidae spp. (Froese and Pauly, 2017). On the other hand, other taxa were not observed either below 50-60 % saturation ("Medium tolerance" pattern), or below 70-80 % saturation ("Low

tolerance" pattern). Taxa following a "Low tolerance" pattern, were mostly reef-associated species, and therefore are not usually seen in mangrove habitats, such as *C. vagabundus*, *H. acuminatus*, and *Scarus* sp. (Froese and Pauly, 2017). The three different types of patterns of utilisation observed may highlight that taxa following a "High tolerance" pattern, and therefore taxa commonly seen in mangrove habitats, are more tolerant to low DO than taxa following "Medium tolerance" and "Low tolerance" patterns.

Even though the underlying adaptations behind these patterns still need to be investigated, these observations suggest that taxa able to withstand low DO were the most successful at using mangrove habitats over taxa displaying avoidance behaviour. Indeed, "High tolerance" taxa were on average more frequently observed than "Medium tolerance" taxa, themselves more frequently observed than "Low tolerance" taxa. Tolerance to low DO provides an evident benefit as



taxa can use mangrove habitats more often and for longer periods compared to taxa that need to migrate temporarily to avoid harmful DO levels. Moreover, remaining in low DO when most other taxa must leave, can provide opportunistic feeding and limited competition (Diaz et al., 1992; Rahel and Nutzman, 1994). On the other hand, tidal migrations can have indirect costs as they can increase risk exposure to predators as fish travel to open water, aggregate fish in suboptimal habitats (less food,

more predation) while waiting for DO conditions to improve, and increase energetic costs during extended swimming activities (Eby et al., 2005; Shoji et al., 2005; Chabot and Claireaux, 2008; Craig, 2012). This implies that being adapted to withstand low DO might be important for taxa using mangrove habitats extensively.

While no differences in DO levels were found between the edge and inside of the forest, it was interesting to note that most taxa venturing in-forest (23 taxa) were following a "High tolerance" pattern (16 taxa; 4 taxa were following a

"Medium tolerance" pattern, and 3 taxa a "Low tolerance" pattern). In other mangrove forests, DO can reach levels close to 0 % saturation (Knight et al., 2013; Mattone and Sheaves, 2017), so it is possible that such lethal levels are also occasionally reached in Bourake. This could explain why relatively few taxa venture inside the forest, and those that do, appear to be highly tolerant to hypoxia.

## 5. Conclusion

The overall value of mangrove forests has been linked to parameters such as geographical location, tidal range (micro-, meso- or macrotidal), setting (coastal, estuarine, island, embayment), and connectivity to adjacent habitats (Unsworth et al., 2008; Igulu et al., 2014; Bradley et al., 2019). While these factors provide important information, this study also shows that for a same mangrove forest, its value is temporally and spatially variable. About half of the variability in fish assemblages was explained by changes in depth, DO, lunar phase, position within the mangrove forest, time of day and tide direction.

Most of the temporal variability occurred on a tidal scale, highlighting the importance of tide in driving mangrove forests utilisation. Here, depth and DO were mostly considered to explain tidal variations in fish assemblages, however, tide can induce variations in many other factors. For instance, a recent study suggested that the resuspension of mangrove-derived organic matter via porewater exchange could temporally boosts primary and secondary production, attracting fish regardless of water quality conditions (David et al., 2018). These results highlight the complexity to quantify the utilisation, and

consequently, the value of mangrove forests and call for more investigations, especially on the effects of tide.

This study is the first to provide insights on how mangrove forests utilisation by fish is influenced by DO. It suggests that tolerance to low DO may be a widespread adaptation for taxa commonly using mangrove forests and could explain why they manage to thrive in these harsh environments. The need of being tolerant to low DO, or able to undertake tidal migrations while limiting alternative costs, is likely to limit the number of taxa using intertidal mangrove habitats that

experience low DO. However, with only field data it is difficult to attribute specific fish responses to DO. Physiological techniques could be used to determine whether a difference in hypoxia tolerance could explain why some species access mangrove habitats at low DO levels while others access at higher DO levels (Lawton, 1991; McGill et al., 2006). This is the



first study to look at, and suggest, a relationship between DO and fish utilisation of mangrove habitats. It adds to our knowledge on factors determining mangrove habitats value and highlights the importance to consider DO as a key controlling factor. More in-depth evaluation of DO dynamics and its impacts on fish populations in other locations would certainly help understanding the heterogeneous value of intertidal mangrove forests.

## 5  Data availability

All relevant data are freely available online on the Tropical Data Hub repository.

DOI: http://doi.org/10.25903/5cd4d312cbcfb

Dubuc, A. (2019): Dataset: fish assemblages and environmental parameters in Bourake. James Cook University. (dataset). http://doi.org/10.25903/5cd4d312cbcfb

## 10  Appendices




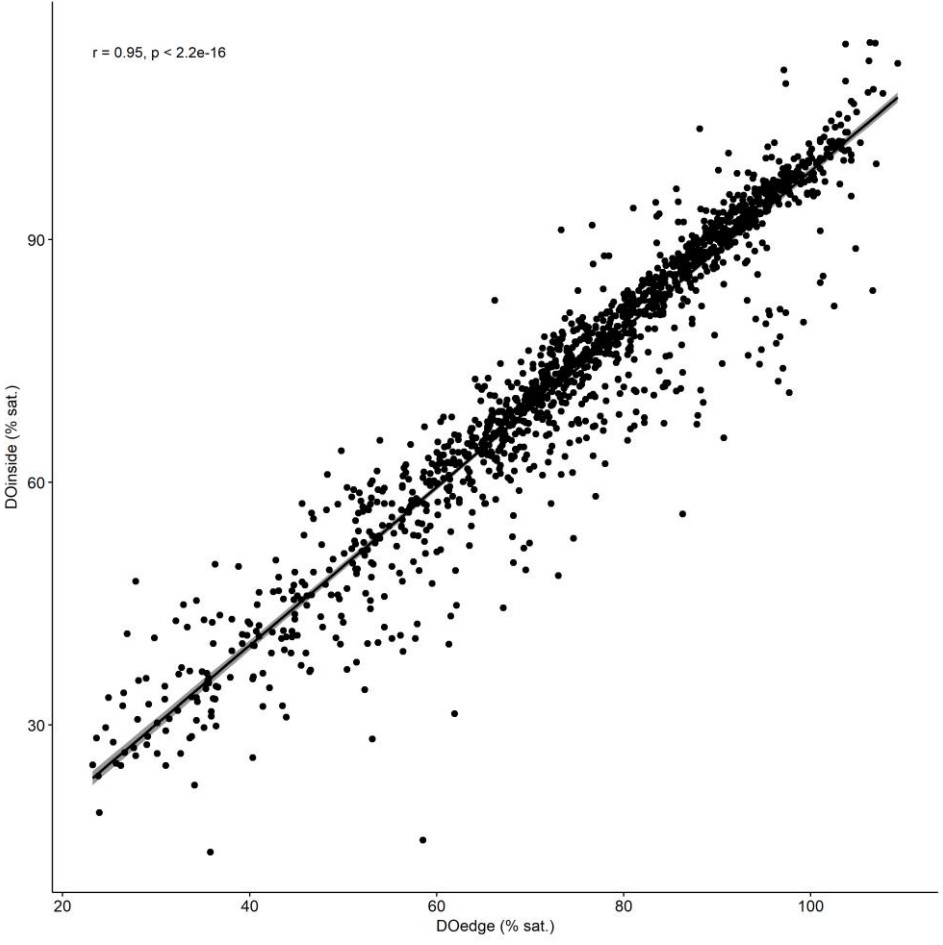

**Figure A1: Kendall's correlation test used to determine whether patterns of change in DO were significantly correlated between edge and in-forest sites.**

**Table A1: The full list of taxa identified by underwater video cameras at Bouraké, New Caledonia.**

| Family | Taxon | Family | Taxon |
|---|---|---|---|
| Acanthuridae | *Acanthurus auranticavus* | Haemulidae | *Plectorhinchus lineatus* |
| | *Acanthurus grammoptilus* | | *Plectorhinchus* spp. |
| | *Acanthurus* sp. cf *blochii* | | *Pomadasys argenteus* |
| | *Ctenochaetus* sp. | Hemiramphidae | *Hyporhamphus* sp. |
| | *Zebrasoma velifer* | Labridae | *Choerodon graphicus* |
| Apogonidae | *Fibramia lateralis* | | Labridae spp. |
| | *Ostorhinchus septemstriatus* | Lethrinidae | *Lethrinus harak* |
| Belonidae | Belonidae spp. | | *Lethrinus lentjan* |
| Blenniidae | Blenniidae spp. | | |



| | | | |
|---|---|---|---|
| Carangidae | *Caranx ignobilis* | | *Lethrinus obsoletus* |
| | *Caranx papuensis* | Lutjanidae | *Lutjanus argentimaculatus* |
| | *Caranx* sp. | | *Lutjanus fulviflamma* |
| Chaetodontidae | *Chaetodon auriga* | | *Lutjanus fulvus* |
| | *Chaetodon bennetti* | | *Lutjanus russellii* |
| | *Chaetodon ephippium* | Monodactylidae | *Monodactylus argenteus* |
| | *Chaetodon flavirostris* | Mugilidae | Mugilidae spp. |
| | *Chaetodon lineolatus* | Mullidae | *Mulloidichthys flavolineatus* |
| | *Chaetodon lunula* | | *Parupeneus ciliatus* |
| | *Chaetodon melannotus* | | *Parupeneus indicus* |
| | *Chaetodon speculum* | | *Upeneus tragula* |
| | *Chaetodon vagabundus* | Pomacanthidae | *Pomacanthus sexstriatus* |
| | *Heniochus acuminatus* | Pomacentridae | *Neopomacentrus* spp. |
| Clupeidae | Clupeidae spp. | Scaridae | *Bolbometopon muricatum* |
| Diodontidae | *Diodon hystrix* | | *Scarus* sp. cf *ghobban* |
| Ephippidae | *Platax pinnatus* | Scatophagidae | *Scatophagus argus* |
| Fistulariidae | *Fistularia* spp. | Serranidae | *Epinephelus caeruleopunctatus* |
| Gerreidae | *Gerres filamentosus* | | *Epinephelus lanceolatus* |
| | *Gerres oyena* | | *Epinephelus malabaricus* |
| Gobiidae | *Amblygobius linki* | | *Epinephelus* sp. |
| | *Amblygobius nocturnus* | Siganidae | *Siganus canaliculatus* |
| | *Amoya gracilis* | | *Siganus lineatus* |
| | *Asterropteryx* sp. cf *striata* | | *Siganus punctatus* |
| | *Cryptocentrus leptocephalus* | Sparidae | *Acanthopagrus* sp. cf *akazakii* |
| | *Eviota* sp. | Sphyraenidae | *Sphyraena barracuda* |
| | *Exyrias puntang* | Tetraodontidae | *Arothron hispidus* |
| | Gobiidae spp. | | |
| | Gobiidae spp.2 | | |
| | *Redigobius balteatus* | | |



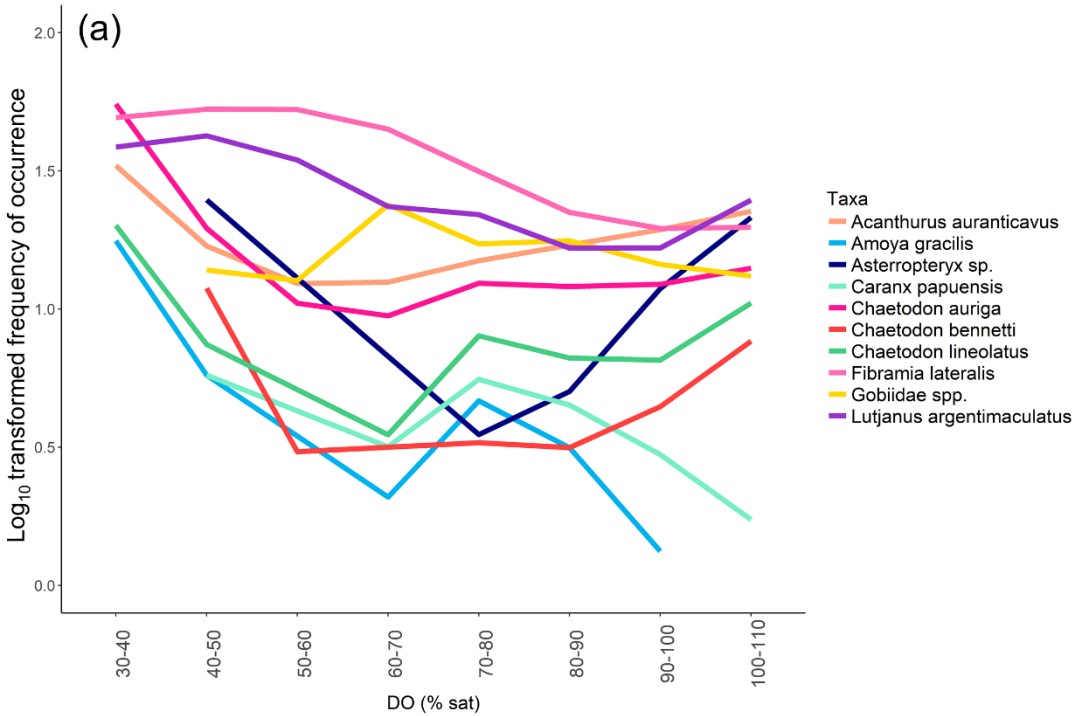

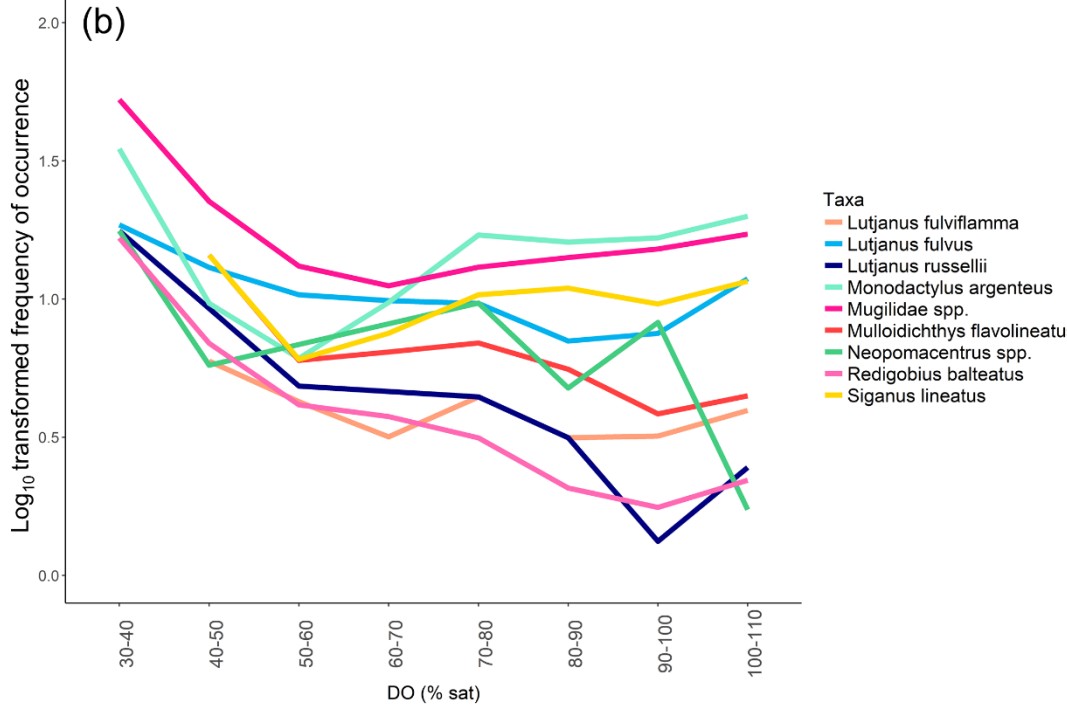





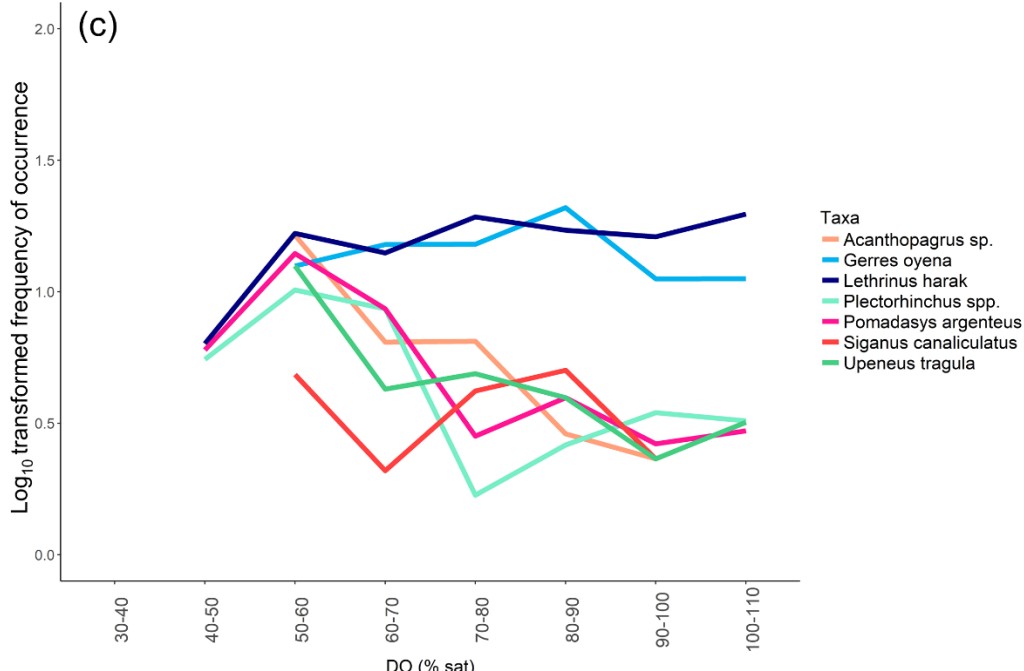

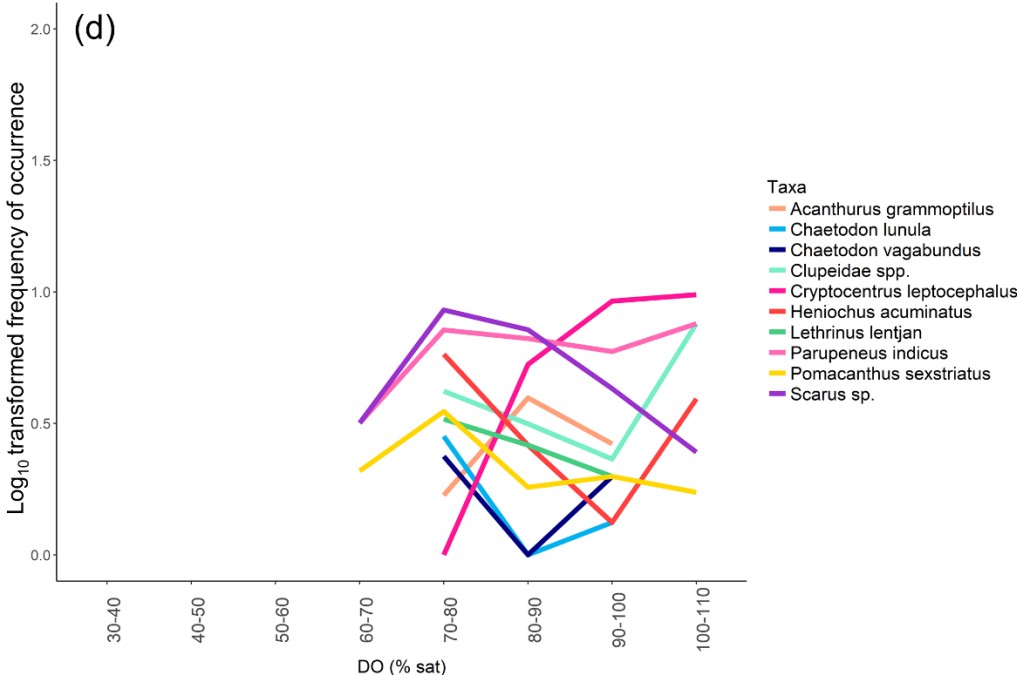

**Figure A2: Species-specific patterns of mangrove utilisation grouped by type of patterns: (a) Pattern 1: "High tolerance; (b) Pattern 1: "High tolerance" (continued); (c) Pattern 2: "Medium tolerance"; (d) Pattern 3: "Low tolerance".**



**Author contribution**

AD, CM, NJW and MS contributed to the conceptualisation. AD led the investigation, project administration, formal analysis, visualisation and writing. CM, NJW, RB, and MS assisted with supervision and writing.

**Competing interests**

The authors declare that they have no conflict of interest.

**Acknowledgements**

We are grateful to Dr. Adrien Jacotot, Dr. Thanh Nho Nguyen and Mrs Clara Hass for their assistance in the field, and to the members of the Science for Integrated Coastal Ecosystem Management group for providing suggestions that improved the
manuscript.

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

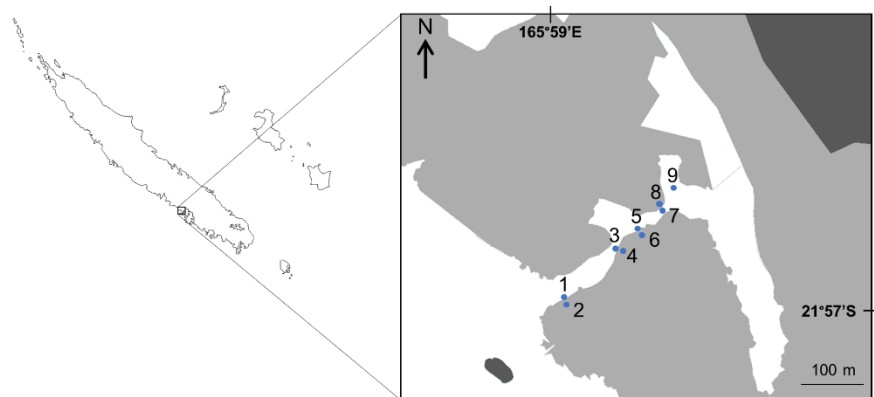

10  **Figure 1: Map of the study system in Bouraké, South Province of New Caledonia. The 9 study sites sampled from 21 February to 1 March 2017 are indicated by their corresponding numbers. Light grey areas represent mangrove forest, dark grey areas represent mainland, and white areas represent water.**





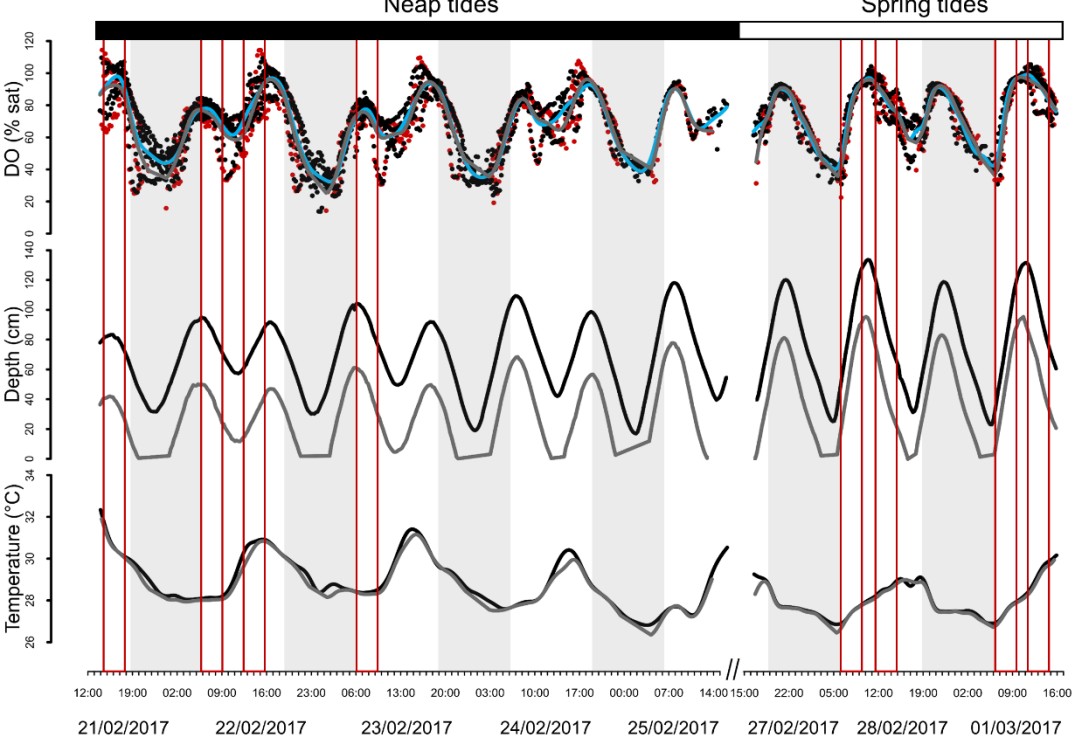

**Figure 2: Cubic spline smoothers for dissolved oxygen (DO), depth and temperature. Data are from 21 February to 25 February 2017 and from 27 February to 1 March 2017. For DO, edge sites are represented by the blue smoother and black points, and in-forest sites by the grey smoother and red points. For the other factors, edge sites are represented by the black smoothers and in-forest sites by the grey smoothers. Shaded areas represent sunset to sunrise. Each red box represents an underwater camera sampling.**





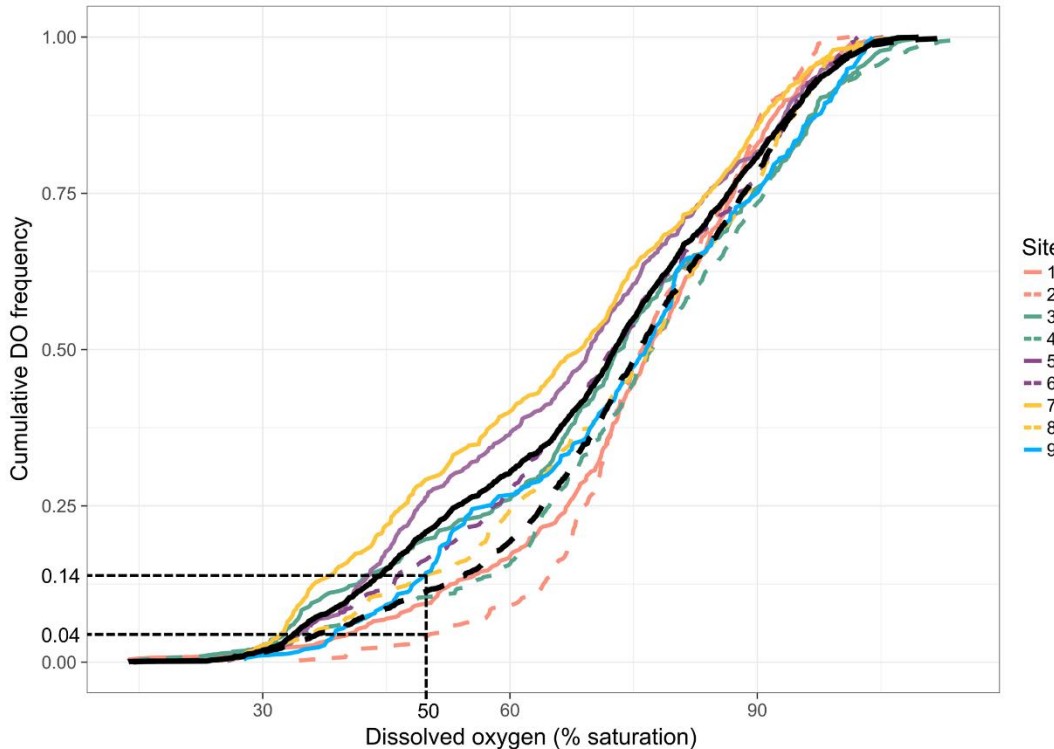

**Figure 3: Site-specific cumulative DO frequencies. Each colour represents a paired site (edge and in-forest), and edge sites are represented by solid coloured lines and in-forest sites by dashed coloured lines. The solid black line indicates the mean cumulative DO frequencies across edge sites and the dashed one the mean cumulative DO frequencies across in-forest sites. The frequency of**
5  **hypoxia (DO ≤ 50 % saturation) at sites 2 and 8 is indicated to help read the figure.**



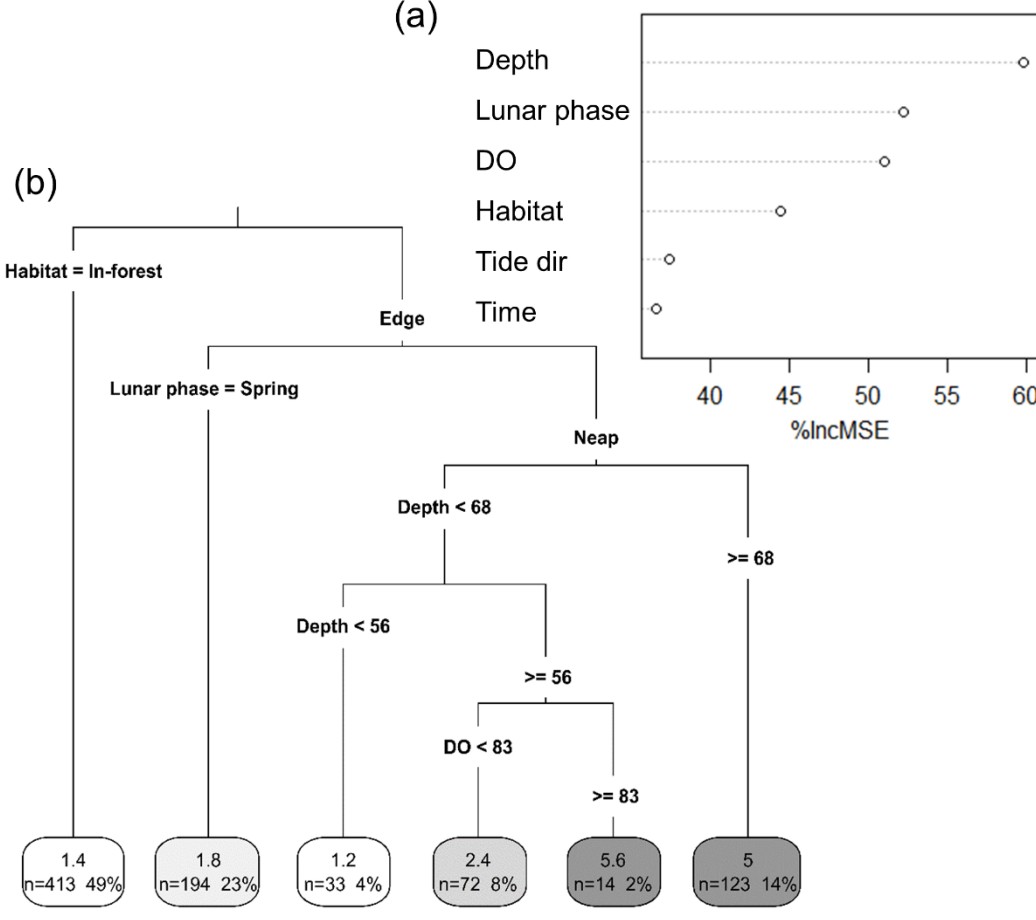

**Figure 4: Importance of different environmental factors in explaining variations in taxonomic richness. (a) Random forest importance plot. Importance plot was obtained from a random forest model built with Site, Depth (cm), Lunar phase (neap vs spring), DO: Dissolved oxygen (% saturation), Time: Time of day (morning vs afternoon), Tide dir: Tide direction (flooding vs ebbing) and Habitat (in-forest vs edge) as predictors for taxonomic richness. (b) Univariate classification tree. The tree was built using the same variables and provides a visual interpretation of the random forest model. Numbers in the boxes in each terminal leaf represent the average taxonomic richness, the number of 5-min intervals (n), and the total % of 5-min intervals that n represents.**





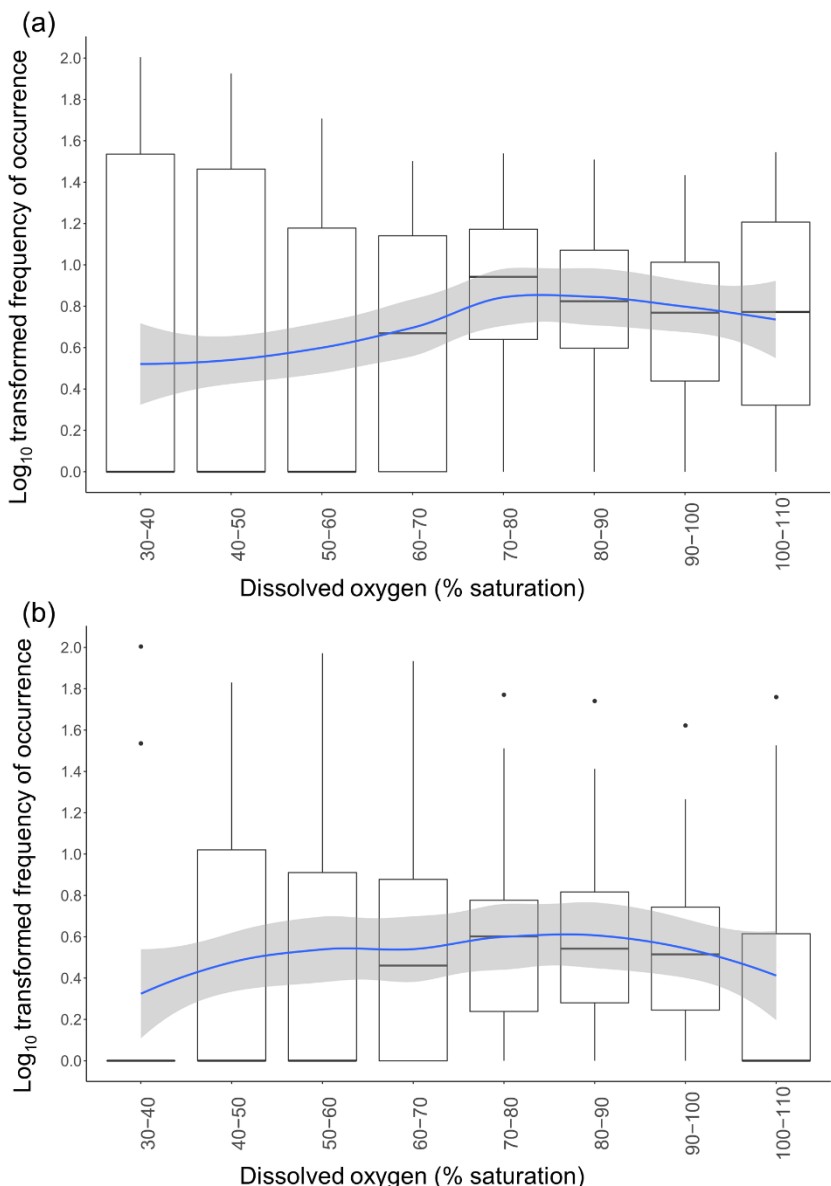

**Figure 5: Variation in frequencies of occurrence of fish across DO class. Frequencies of occurrence were log₁₀ transformed. Each data point used to draw the boxplots represents the frequency of occurrence of one common taxon during a specific DO class. The blue line represents the GAMM model fitted with DO as the smooth term using a Gaussian distribution and an identity link function for (a) Edge sites; and (b) In-forest sites. Shaded areas represent the confidence interval at 95 %.**





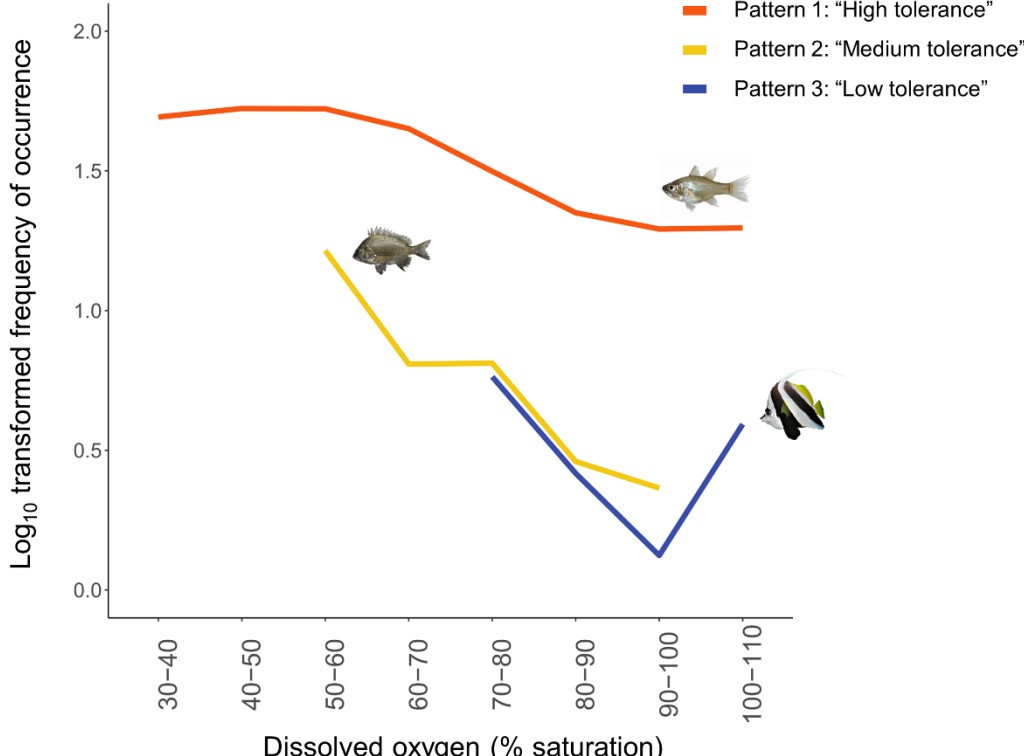

**Figure 6: The 3 common patterns of mangrove utilisation across DO identified.** Each LOESS curve represents one example of taxa per type of patterns of mangrove utilisation across DO: Pattern 1: "High tolerance" represented by taxon *Fibramia lateralis*; 2) Pattern 2: "Medium tolerance" represented by taxon *Acanthopagrus* sp.; 3) Pattern 3: "Low tolerance" represented by taxon *Heniochus acuminatus*. LOESS curves were built with the $\log_{10}$ transformed frequencies of occurrence.





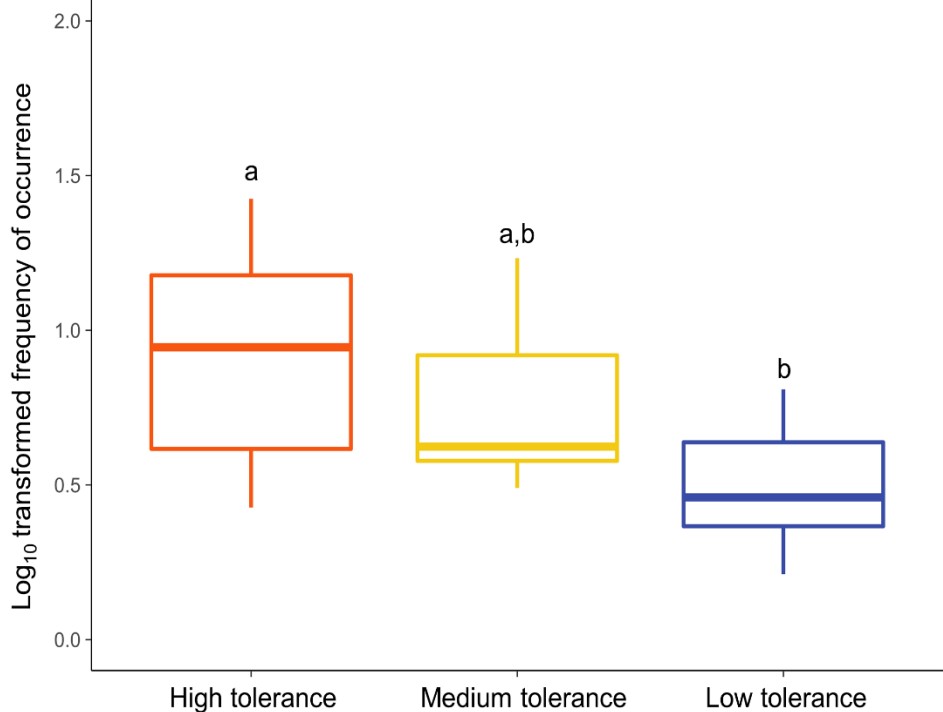

**Figure 7: Relationship between frequencies of occurrence and type of patterns followed. Overall frequencies of occurrence were calculated for each common taxon at the DO range corresponding to that taxon's assigned pattern of utilisation. Differential letters above boxes denote statistically different means of frequency of occurrence among types of patterns of utilisation (Dunn test: p < 0.05).**

**Table 1: Summary of the environmental factors during the study period. For each factor, the minimum, maximum and mean (± SE) values are provided for neap tides and spring tides, on the edge and in-forest sites.**

| Environmental factors | Values | Neap | | Spring | |
|---|---|---|---|---|---|
| | | Edge | In-forest | Edge | In-forest |
| DO (% saturation) | Min | 13.7 | 14.3 | 30.9 | 22.6 |
| | Max | 110.6 | 114.4 | 105.5 | 103.3 |
| | Mean (± SE) | 67.2 (± 0.6) | 71.4 (± 0.6) | 76.8 (± 0.7) | 79.5 (± 0.7) |
| Temperature (°C) | Min | 26.2 | 25.8 | 25.9 | 25.1 |
| | Max | 32.0 | 31.5 | 30.5 | 30.4 |
| | Mean (± SE) | 29.1 (± 0.0) | 28.9 (± 0.0) | 28.0 (± 0.0) | 28.0 (± 0.0) |
| Water depth (cm) | Min | 1.1 | 0.0 | 2.4 | 0.0 |
| | Max | 118.1 | 77.8 | 133.7 | 95.5 |
| | Mean (± SE) | 55 (± 0.7) | 34 (± 0.6) | 71 (± 1.2) | 48 (± 1.1) |
| Tidal range (m) | Min | 0.35 | 0.35 | 1.25 | 1.25 |
| | Max | 0.59 | 0.59 | 1.38 | 1.38 |



| | Mean | 0.46 | 0.46 | 1.31 | 1.31 |
|---|---|---|---|---|---|

**Table 2: The 36 common fish taxa identified by underwater video cameras at Bouraké, New Caledonia. The superscript number corresponds to the type of patterns of mangrove utilisation across DO followed by the taxon (Fig. 6): Pattern 1: "High tolerance"; Pattern 2: "Medium tolerance"; Pattern 3: "Low tolerance". Taxa highlighted in bold represent the 10 most common taxa. Taxa recorded in-forest (5 m inside the forest) are underlined.**

| Family | Taxon | Family | Taxon |
|---|---|---|---|
| Acanthuridae | ***Acanthurus auranticavus*** [1] | Lethrinidae | ***Lethrinus harak*** [2] |
| | *Acanthurus grammoptilus* [3] | | *Lethrinus lentjan* [3] |
| Apogonidae | ***Fibramia lateralis*** [1] | Lutjanidae | ***Lutjanus argentimaculatus*** [1] |
| Carangidae | *Caranx papuensis* [1] | | *Lutjanus fulviflamma* [1] |
| Chaetodontidae | ***Chaetodon auriga*** [1] | | *Lutjanus fulvus* [1] |
| | *Chaetodon bennetti* [1] | | *Lutjanus russellii* [1] |
| | *Chaetodon lineolatus* [1] | Monodactylidae | ***Monodactylus argenteus*** [1] |
| | *Chaetodon lunula* [3] | Mugilidae | **Mugilidae spp.** [1] |
| | *Chaetodon vagabundus* [3] | Mullidae | *Mulloidichthys flavolineatus* [1] |
| | *Heniochus acuminatus* [3] | | *Parupeneus indicus* [3] |
| Clupeidae | *Clupeidae spp.* [3] | | *Upeneus tragula* [2] |
| Gerreidae | ***Gerres oyena*** [2] | Pomacanthidae | *Pomacanthus sexstriatus* [3] |
| Gobiidae | *Amoya gracilis* [1] | Pomacentridae | *Neopomacentrus* spp. [1] |
| | *Asterropteryx* sp. cf *striata* [1] | Scaridae | *Scarus* sp. cf *ghobban* [3] |
| | *Cryptocentrus leptocephalus* [3] | Siganidae | *Siganus canaliculatus* [2] |
| | **Gobiidae spp.** [1] | | ***Siganus lineatus*** [1] |
| | *Redigobius balteatus* [1] | Sparidae | *Acanthopagrus* sp. cf *akazakii* [2] |
| Haemulidae | *Plectorhinchus* spp. [2] | | |
| | *Pomadasys argenteus* [2] | | |