# Peer review of "Hypoxia in mangroves: occurrence and impact on valuable tropical fish habitat"

_Biogeosciences, 2019_

## Referee Comment (RC1) · Keryn Gedan (Referee) · 21 Jun 2019

Overall, I think this is a fantastic observational dataset of variation in fish communities during tidally driven variation in DO. The DO data themselves, sampled across a spatial gradient in the mangroves, are very interesting and nicely plotted. The authors recorded fish videos at an impressive number of sites and hours in this remote location, and identified a large number of fish species. The analysis is creative.

Spatial patterns in DO within the mangrove forest: p. 12, Line 8 notes no difference between in-forest and edge sites, however in the results, p. 7, Line 20 describes a difference in the frequency of DO <50% saturation in-forest and edge environments, with low DO observed more frequently in edge (channel) sites than in-forest. This

result surprised me, given that that the in-forest sites would have less water exchange than edge sites. Was this a true difference or not? If there was truly no difference, why do you think the expected pattern of lower DO in the forest than the channel was not observed?

Discussion section 4.1 "Tidal migrations: stranding or hypoxia?": The text does not answer the question in the section title. Low DO was correlated with the tidal cycle, so it is impossible to disentangle fish movements in response to tidal variation/depth relative to DO. This is carefully worded in the introduction, but not the abstract, which suggests that fish migrate due to low DO, rather than DO being a factor that may drive tidal migrations. The first discussion section would be stronger with a different section title.

Related, did in-forest sites bottom out during low tides? It looks as though they did in Fig. 2, for example, just after an ebb tide video period on 2/28/17. This would be an opportunity to investigate stranding patterns. Did all fish leave the site towards the end of this falling tide? That particular occasion was a relatively mild DO period (40-80% saturation), and so perhaps the effects of shallow water and low DO could be separated, at least anecdotally, if not statistically, on that date.

Discussion section 4.2 "Tidal-induced dissolved oxygen variations" presents two processes that reduce DO in mangrove waters: 1) diel cycles that are the result of dominance of photosynthesis effects during daylight hours and respiration at night, and 2) tidal pumping of porewater into the water column on ebbing tides. These two processes should be better distinguished within the introduction and discussion sections. Separating discussion of the two factors and their relevance to this study (for which fish data is daytime-only) into separate paragraphs would be helpful.

The discussion section on "4.3 Species-specific responses to DO variations" highlights some very important results from this study. However, there is a missed opportunity that no species are specifically discussed. What can be said about the fish species,

genera, or families observed in the high, medium, and low tolerance behavioral groups identified?

Minor comments

Please add a legend to the map in Fig. 1.

p. 3, Line 6, "implying that mangrove forests are especially vulnerable to anthropogenic deoxygenation due to their location along the coasts" – I would say, "the aquatic communities within mangrove forests" are vulnerable, since the mangrove forests themselves are fairly resistant to deoxygenation as low DO does not negatively affect the foundation species of mangroves, and the forest itself can withstand low DO stress quite well.

p. 4, Line 28, What does VLC stand for?

p. 5, Line 20, In the random forest model, what happens in the case of co-linear variables? Also, can RF account for interactions between predictors?

p. 5, Line 30, Please explain "out-of-bag error."

p. 10, Line 16, Tidal period is generally defined as high to high or low to low. High to low would be half of a period.

p. 11, Line 21, parenthetical "High tolerance pattern" is confusing. I suggest "(i.e. most species exhibited the "High tolerance" pattern)."

p. 12, Line 15, it seems like "value" of mangroves, here, implies that fish diversity or abundance is a proxy for mangrove forest value? Please clarify.

---

## Referee Comment (RC2) · Anonymous Referee #2 · 10 Aug 2019

Overall, the paper is well written and deals with an important gap in current knowledge around utilization patterns of mangrove habitats by fish species. The material and methods section is really impressive and authors provide sufficient results to support their conclusions. There are only a few clarifications that need to be done.

The methods section is really detailed and informative, but a few questions remains. For example, how could the correlation between DO and Depth affect the RF model? In the results section authors state that different models were created for each variable, but it is still unclear how the model is affected by this correlation.

Also, in results, lunar phase was the second most important variable in explaining species richness, and although it seems to be an important result, there is no further discussion on this.

[Figure]

In results (page 9) and discussion 4.3 section (page 11) authors state that tolerance groups were represented by different groups of species (with reef associated taxa having low tolerance behavior). However, this statement is not further explored. Resident and vagrant fish are likely to differ in the way they use mangrove habitats, so, how DO may affect their patterns of utilization?

The discussion section ends with the following statement "This could explain why relatively few taxa venture inside the forest, and those that do, appear to be highly tolerant to hypoxia" (page 11), but there is no previous information on spatial distribution of fish species between both environments. Moreover, although no differences in DO was found between in forest and edge samples, what about the other environmental variables? Looking at table 1, it seems like water depth may differ between both sites, which could explain the lower species richness.

---

## Author Comment (AC1) · 30 Aug 2019

**Response to Keryn Gedan (referee 1)**

**General comment**
Overall, I think this is a fantastic observational dataset of variation in fish communities during tidally driven variation in DO. The DO data themselves, sampled across a spatial gradient in the mangroves, are very interesting and nicely plotted. The authors recorded fish videos at an impressive number of sites and hours in this remote location, and identified a large number of fish species. The analysis is creative.

**Response**
We thank Keryn Gedan for her insightful comments that improved the manuscript. We carefully addressed each comment below.

**Comment #1**
Spatial patterns in DO within the mangrove forest: p. 12, Line 8 notes no difference between in-forest and edge sites, however in the results, p. 7, Line 20 describes a difference in the frequency of DO <50% saturation in-forest and edge environments, with low DO observed more frequently in edge (channel) sites than in-forest. This result surprised me, given that that the in-forest sites would have less water exchange than edge sites. Was this a true difference or not? If there was truly no difference, why do you think the expected pattern of lower DO in the forest than the channel was not observed?

**Response**
This is a very interesting point that surprised me as well. First of all, the difference in mean DO is not significant between edge and in-forest. The slightly lower DO levels found on the edge I believe are due to the fact that water decreases too fast compare to DO in-forest, therefore within the time that the in-forest is inundated, DO does not decline as much as on the edge where very shallow water remains permanently as it does not get exposed at low tide. This sentence has been added to the discussion to make this point: "There was no difference observed between DO dynamics on the edge and in-forest, however minimum values were slightly lower on the edge because water remained permanently at low tide being subjected to further decline compare to in-forest that became exposed earlier during the tide and therefore experiencing a shorter DO decline period." p.11, lines 17-20.

**Comment #2**
Discussion section 4.1 "Tidal migrations: stranding or hypoxia?": The text does not answer the question in the section title. Low DO was correlated with the tidal cycle, so it is impossible to disentangle fish movements in response to tidal variation/depth relative to DO. This is carefully worded in the introduction, but not the abstract, which suggests that fish migrate due to low DO, rather than DO being a factor that may drive tidal migrations. The first discussion section would be stronger with a different section title.

**Response**
The section title has been modified as suggested to be more accurate: "Depth and DO are both potential factors for observed tidal migrations" p.9, line 32.

**Comment #3**
Related, did in-forest sites bottom out during low tides? It looks as though they did in Fig. 2, for example, just after an ebb tide video period on 2/28/17. This would be an opportunity to investigate stranding patterns. Did all fish leave the site towards the end of this falling tide? That particular occasion was a relatively mild DO period (40-80% saturation), and so perhaps the effects of shallow water and low DO could be separated, at least anecdotally, if not statistically, on that date.

**Response**
This is a great idea! Unfortunately, I did not sample until the end of the falling tide this day, therefore I do not have the data to identify whether fish were still present towards the end of the ebbing tide. But a future targeted sampling at the end of ebbing tide could indeed provide interesting results to help disentangle the effect of low DO and shallow water depth. In Dubuc et al. (2019) we studied more specifically fish responses to depth patterns and showed that certain species did not use mangrove habitats while depth was high enough for fish to safely access, emphasising that water quality, and especially DO, is probably an important parameter to take into consideration for habitat utilisation.

**Comment #4**

Discussion section 4.2 "Tidal-induced dissolved oxygen variations" presents two processes that reduce DO in mangrove waters: 1) diel cycles that are the result of dominance of photosynthesis effects during daylight hours and respiration at night, and 2) tidal pumping of porewater into the water column on ebbing tides. These two processes should be better distinguished within the introduction and discussion sections. Separating discussion of the two factors and their relevance to this study (for which fish data is daytime-only) into separate paragraphs would be helpful.

**Response**
In the introduction, a brief explanation on how DO fluctuates in response to the autotrophic cycle and tide has been added: "The main factor responsible for DO fluctuations is the autotrophic cycle, with photosynthesis occurring during daylight hours, and respiration during nighttime hour, creating a diel-cycle in DO. Another important parameter to consider, especially in intertidal environments, is tide as it is responsible for many physical and chemical changes susceptible to impact oxygen cycle. If these two factors are considered, DO ca be partially predicted, with the lowest DO levels occurring at night or dawn at low tide, following nighttime respiration, while maximum levels are recorded in the afternoon at high tide, following autotrophic production (Kenney et al., 1988; Mazda et al., 1990; D'Avanzo and Kremer, 1994; Tyler et al., 2009)." p.2, lines 25-30. The manuscript has been modified to clarify this point, especially the fact that fish assemblages were only collected during daytime, and therefore, part of the answer on how DO diel fluctuations impact fish assemblages is not known. p.11, lines 22-25.

**Comment #5**
The discussion section on "4.3 Species-specific responses to DO variations" highlights some very important results from this study. However, there is a missed opportunity that no species are specifically discussed. What can be said about the fish species, genera, or families observed in the high, medium, and low tolerance behavioral groups identified?

**Response**
This is an important result from the study indeed, and this discussion section has been extended and modified to emphasise it: section 4.3 p.11-12. It was interesting to notice that species observed in high tolerance group were all species commonly associated with mangrove habitats, using them extensively, while species from the other two groups are less commonly observed, even never recorded before in mangrove habitats.

**Comment #6**
Please add a legend to the map in Fig. 1.

**Response**
A legend has been added as suggested.

**Comment #7**
p. 3, Line 6, "implying that mangrove forests are especially vulnerable to anthropogenic deoxygenation due to their location along the coasts" – I would say, "the aquatic communities within mangrove forests" are vulnerable, since the mangrove forests themselves are fairly resistant to deoxygenation as low DO does not negatively affect the foundation species of mangroves, and the forest itself can withstand low DO stress quite well.

**Response**
The sentence has been modified as follow to address this unprecise language issues: "implying that mangrove ecosystems are especially prone to experience anthropogenic deoxygenation due to their location along the coasts." p.3, lines 8-9.

**Comment #8**
p. 4, Line 28, What does VLC stand for?

**Response**
The acronym VLC has been defined in the manuscript to address this comment p.4, line 28.

**Comment #9**
p. 5, Line 20, In the random forest model, what happens in the case of co-linear variables? Also, can RF account for interactions between predictors?

**Response**

There is no agreement on how correlated variables impact the prediction of Random Forest models (Grömping, 2009; Neville, 2013). However random forest is a very robust method, and in this study, considering the large dataset used with only few predictors that are all relevant to explain fish assemblages, we believe that overfitting is not an issue. I am not aware of any coding available to account for interactions between predictors in random forest models, but a recent paper has suggested some that could be made available in the future (Gregorutti et al., 2017). Here, we tested for the effect of correlation by building two models, one excluding depth, and the other one excluding DO, to confirm that both variables increased substantially the prediction capacity of the random forest, which was the case.

**Comment #10**

p. 5, Line 30, Please explain "out-of-bag error."

**Response**

A short definition of "out-of-bag error" has been added to the manuscript: p.9, lines 30-31.

**Comment #11**

p. 10, Line 16, Tidal period is generally defined as high to high or low to low. High to low would be half of a period.

**Response**

The sentence has been modified to use the correct term as advised p.10, line 18.

**Comment #12**

p. 11, Line 21, parenthetical "High tolerance pattern" is confusing. I suggest "(i.e. most species exhibited the "High tolerance" pattern)."

**Response**

The sentence has been modified to enhance understanding as suggested: p.11, line 28-29.

**Comment #13**

p. 12, Line 15, it seems like "value" of mangroves, here, implies that fish diversity or abundance is a proxy for mangrove forest value? Please clarify.

**Response**

To avoid confusion around the term "value" a definition has been added when first mentioned in the introduction p.1, line 14; p.1, line 31.

---

## Author Comment (AC2) · 30 Aug 2019

**Response to referee 2**

**General comment**
Overall, the paper is well written and deals with an important gap in current knowledge around utilization patterns of mangrove habitats by fish species. The material and methods section is really impressive and authors provide sufficient results to support their conclusions. There are only a few clarifications that need to be done.

**Response**
We thank referee #2 for insightful comments that improved the manuscript. We carefully addressed each comment below.

**Comment #1**
For example, how could the correlation between DO and Depth affect the RF model? In the results section authors state that different models were created for each variable, but it is still unclear how the model is affected by this correlation.

**Response**
There is no agreement on how correlated variables impact the prediction of Random Forest models (Grömping, 2009; Neville, 2013). However random forest is a very robust method, and in this study, considering the large dataset used with only few predictors that are all relevant to explain fish assemblages, we believe that overfitting is not an issue. Moreover, the two different models created, one excluding depth and one excluding DO, confirmed that both variables substantially increased the prediction capacity of the model.

**Comment #2**
Also, in results, lunar phase was the second most important variable in explaining species richness, and although it seems to be an important result, there is no further discussion on this.

**Response**
The effect of lunar phase on fish assemblages has been discussed in detail in Dubuc et al. (2019), therefore the discussion of this paper focuses on water depth and DO, that are two factors potentially responsible for the main tidal variation pattern identified in fish assemblages. A sentence has been added at the beginning of the relevant discussion section to clarify this point: "The effects of lunar phase and location on fish assemblages were investigated in detail in Dubuc et al. (2019), therefore the following discussion focuses on water depth and DO, that both varied at a tidal scale.", p.10 lines 2-3.

**Comment #3**
In results (page 9) and discussion 4.3 section (page 11) authors state that tolerance groups were represented by different groups of species (with reef associated taxa having low tolerance behavior). However, this statement is not further explored. Resident and vagrant fish are likely to differ in the way they use mangrove habitats, so, how DO may affect their patterns of utilization?

**Response**
This is an important result, and this discussion section has been extended and modified to emphasise it: section 4.3 p.11-12. We believe that species able to extensively use mangroves are able to do so partly because they are highly tolerant to hypoxia and probably highly tolerant to environmental stressors in general.

**Comment #4**
The discussion section ends with the following statement "This could explain why relatively few taxa venture inside the forest, and those that do, appear to be highly tolerant to hypoxia" (page 11), but there is no previous information on spatial distribution of fish species between both environments. Moreover, although no differences in DO was found between in forest and edge samples, what about the other environmental variables? Looking at table 1, it seems like water depth may differ between both sites, which could explain the lower species richness.

**Response**
A result section has been added to highlight the difference in fish taxonomic richness between the in-forest and the edge, showing that less taxa were observed in-forest: p.7, lines 33-34. Relevant references have also been added to support this point: p.11, line 31. We agree with your comment, and indeed differences in water depth, especially the fact that in-forest sites get exposed at low tide while edge sites remain submerged, could explain differences in taxonomic richness. This point was thoughtfully discussed in another paper (Dubuc et al., 2019).

The relevant paragraph in the discussion has been modified to provide more information regarding this point: p.12, lines 20-27.

---

## Editor Comment (EC1) · Kenneth Rose (Editor) · 3 Sep 2019

There are several author responses to reviewers' comments that are unclear as to what, if any, changes will be made to a revised version of the manuscript. Please review all responses to ensure that what actual changes will be made to manuscript are very clearly stated. In particular, the responses to Comment #3 and Comment #9 of Reviewer 1, and Comment #1 of Reviewer 2 need to be clarified. Note that Comment #9 of reviewer 1 is very similar to Comment #1 of Reviewer 2 and so should be addressed in the revised manuscript. The response text to these comments comes close to what needs to be added to the manuscript.

---

## Author Comment (AC3) · 5 Sep 2019

Dear Prof Rose,

We apologise if our responses were unclear as to what changes have been made. We carefully addressed each reviewer's comments and made the necessary changes to the manuscript. Please find below more details on the changes that have been made, especially regarding comments #3 and #9 of reviewer 1 and comment #1 of reviewer 2.

Reviewer 1:

Comment #3

Related, did in-forest sites bottom out during low tides? It looks as though they did in Fig. 2, for example, just after an ebb tide video period on 2/28/17. This would be an opportunity to investigate stranding patterns. Did all fish leave the site towards the end of this falling tide? That particular occasion was a relatively mild DO period (40-80% saturation), and so perhaps the effects of shallow water and low DO could be separated, at least anecdotally, if not statistically, on that date.

Response

To complete our response, the following sentence has been added to the manuscript to highlight this interesting suggestion: "Future targeted sampling could for instance specifically investigate fish movements at the end of ebbing tides experiencing relatively high DO and low DO, as on the 28 February afternoon and morning respectively. This would help to determine whether fish consistently leave mangrove habitats at a same depth or whether responses are indeed affected by DO levels abnormally low or high.", p.11, lines 3-6.

Comment #9

p. 5, Line 20, In the random forest model, what happens in the case of co-linear variables? Also, can RF account for interactions between predictors?

Response

To complete our response, the following sentence has been added to show that this issue was investigated and addressed: "DO and depth were highly correlated, which can potentially impact the RF prediction of variable importance, although there is no agreement on what the effects of multicollinearity are (Gregorutti et al., 2017). However, RF is a very robust method, and in this study, considering the large dataset used with only few predictors that are all relevant to explain fish assemblages, we believe that overfitting is not an issue. Nevertheless, a RF model was built only with "Depth", and then only with "DO" to test for the effect of multicollinearity on their relative importance.", p.8,

lines 9-14.

Reviewer 2:

Comment #1

For example, how could the correlation between DO and Depth affect the RF model? In the results section authors state that different models were created for each variable, but it is still unclear how the model is affected by this correlation.

Response

To complete our response, the following sentence has been added to show that this issue was investigated and addressed: "DO and depth were highly correlated, which can potentially impact the RF prediction of variable importance, although there is no agreement on what the effects of multicollinearity are (Gregorutti et al., 2017). However, RF is a very robust method, and in this study, considering the large dataset used with only few predictors that are all relevant to explain fish assemblages, we believe that overfitting is not an issue. Nevertheless, a RF model was built only with "Depth", and then only with "DO" to test for the effect of multicollinearity on their relative importance.", p.8, lines 9-14.